# Cytokine Storm—Definition, Causes, and Implications

**DOI:** 10.3390/ijms231911740

**Published:** 2022-10-03

**Authors:** Dominik Jarczak, Axel Nierhaus

**Affiliations:** Department of Intensive Care Medicine, University Medical Center Hamburg-Eppendorf, 20251 Hamburg, Germany

**Keywords:** cytokine storm, cytokine release syndrome, immunity, sepsis, post-cardiac arrest syndrome, overwhelming post-splenectomy infection, CAR-T cell therapy, invasive meningococcal disease

## Abstract

The human innate and adaptive immune systems consist of effector cells producing cytokines (interleukins, interferons, chemokines, and numerous other mediators). Usually, a fragile equilibrium of pro- and anti-inflammation effects is maintained by complex regulatory mechanisms. Disturbances of this homeostasis can lead to intricate chain reactions resulting in a massive release of cytokines. This may result in a drastic self-reinforcement of various feedback mechanisms, which can ultimately lead to systemic damage, multi-organ failure, or death. Not only pathogens can initiate such disturbances, but also congenital diseases or immunomodulatory therapies. Due to the complex and diverse interactions within the innate and adaptive immune systems, the understanding of this important clinical syndrome is incomplete to date and effective therapeutic approaches remain scarce.

## 1. Introduction

Various pathogens, autoimmune and malignant diseases, but also genetic disorders and certain therapeutic interventions, can lead to life-threatening systemic inflammatory syndromes in the human body. Their common feature is a massive release of cytokines due to excessive activation of immune cells. This dysregulated inflammatory response leads to self-reinforcing feedback and may, ultimately, be life-threatening to the host.

These conditions are widely referred to as cytokine release syndrome (CRS) or, in particularly severe courses, cytokine storm (CS). The term CS was first used by James L. Ferrara in 1993 to describe acute graft-versus-host disease (GvHD) in the setting of engraftment syndrome following allogeneic stem-cell transplantation [1]. The term CRS originated with L. Chatenoud, who used it in 1991 to describe a muromonab-induced anti-CD3 syndrome in the setting of immunosuppressive therapy in solid organ transplantation [2].

CS occurs frequently in the context of certain diseases, syndromes, and therapies, for example, anaphylaxis, graft-versus-host disease, acute respiratory distress syndrome (ARDS), and systemic inflammatory response syndrome (SIRS), as well as chimeric antigen receptor-T (CAR-T) cell therapy and sepsis—the latter accounting for up to 19.7% of all deaths worldwide [3].

To date, there is no valid definition for the term CS. It is usually understood to mean an overwhelming immune response characterized by the release of cytokines, including interleukins, interferons, chemokines, and other mediators (see Table 1). These mediators are part of an evolutionarily well-conserved innate immune response that is required for efficient elimination of infectious agents and the repair processes that immediately follow [4].

CS means that the dynamics and quantity of systemically released cytokines cause serious damage in the host organism. However, distinguishing between an appropriate and a pathologically dysregulated inflammatory response in critical illness is difficult or impossible [5]. Since most of the mediators involved in the CS exhibit pleiotropic downstream effects and, in addition, are often interdependent in their biological activity, an extremely complex dynamic arises [6]. The interaction of the mediators and the signaling pathways triggered by them are neither linear nor uniform. Moreover, their quantitative values may indicate the severity of the reactions, but not necessarily pathogenesis, clinical features, and prognosis (see Table 2). This complex interplay highlights the limitations of intervening in the acute inflammatory response based on single mediators and at undifferentiated time points. From a practical point of view, the measurement/detection of elevated cytokines currently almost exclusively refers to the bloodstream. The leading marker is interleukin (IL)-6, although even today this parameter cannot be determined in all centers, let alone other cytokines, which are also elevated [7,8]. Further, there is evidence that both the composition and the quantity of different cytokines vary significantly in different compartments. This has been demonstrated for ascites, broncho-alveolar fluid, pleural effusions, lymph, and urine. However, these compartments are not routinely accessible, and blood concentrations may well represent “the tip of the iceberg” [9,10].

CS and the resulting systemic response can progress from non-specific physical symptoms to multi-organ failure if identified too late and treated inadequately [26]. Clinically, most patients develop febrile temperatures at the onset of CS, which may progress to high fever in severe courses. Other common symptoms in the early phase may include headache, diarrhea, fatigue, rash, arthralgias, and myalgias; neuropsychiatric changes (“septic encephalopathy”) are also common.

Depending on the underlying causes and therapeutic measures, cases of CS differ from each other both in onset and duration [27]. However, the longer the clinical course lasts, the more similar the courses become, so that with advanced progression, there is almost a uniform clinical picture—regardless of the original triggers.

From a clinical point of view, the earliest possible detection of excessive cytokine release is extremely important, as it may be associated with therapeutic decisions and, ultimately, with prognosis and outcome [28]. Sepsis pathology is complex. Cytokine composition changes over time, and anti-inflammatory mediators (e.g., IL-4, IL-10, IL-13, anti-IL-1ra) are present already at the beginning of inflammation. In patients with pre-existing immunological impairments (chronic diseases or iatrogenic), the early proinflammation may even be lacking [29].

However, there is an interdependence of the concentrations of pro- and anti-inflammatory cytokines. The more severe the proinflammatory response expressed, the higher the anti-inflammatory levels will be, and this may lead to a net effect of significant immunosuppression. The theory of two sequential phases of the host immune response, where a primary phase of hyperinflammation is followed by a “compensatory” anti-inflammatory response, leading to immunosuppression, had to be abandoned. Rather, a concept of simultaneous activation of pro- and anti-inflammatory responses is generally accepted [30,31].

The complexity of the sepsis syndrome, together with the multi-level inter-organ cross talk, entails that even after more than three decades of research, there is no specific cure—and there probably never will be. Even in precision or personalized medicine, where treatments are targeted at pre-specified conditions or individual patient requirements, no ground-breaking success has been achieved to date. An example of this is the approach known as theranostics, where selected biomarkers are used to choose a specific therapy and simultaneously measure the response to this treatment [32]. However, it is likely that there is a right time for each element of the immune response to be enhanced or attenuated during the defense against severe infections. Initially, when the pathogen load is high, the demands on the immune system are quite different from those at a later stage, when the pathogens are largely contained by effective anti-infective measures. In other words, applying the right therapeutic approach at the wrong time can potentially worsen clinical outcomes [6]. Furthermore, if anti-inflammatory interventions are only effective in certain subgroups of patients, demonstrating their efficacy in heterogeneous populations recruited in most clinical sepsis trials remains extremely difficult. Hence, there is an urgent need for well-designed therapeutic trials—otherwise, the potential benefit simply cannot be worked out due to a poor signal-to-noise ratio [33]. 

In general, infection as the cause of CS must be excluded early and reliably, and the function of important organ systems must be assessed based on laboratory chemistry parameters. If an infectious cause can be ruled out, CS can be identified based on repeatedly and profoundly elevated cytokine levels. That being said, it is difficult to clearly distinguish a high-grade inflammatory response from a dysregulated host response in severe infection. Profiles of different cytokines (e.g., IL-6/IL-10 ratio) can be helpful in identifying a trend for the further course based on baseline values [34]. However, they are mostly not available in a timely manner and are of limited use in making prompt treatment decisions.

## 2. Pathophysiology of CS

Inflammation is the mechanism that multicellular organisms have evolved to defeat invasive pathogens and initiate healing of injured tissue. A balanced, “protective” inflammatory response consists of diverse mechanisms and involves activation of both pro- and anti-inflammatory pathways within the innate and the acquired immune systems [4]. The immune system can recognize and counteract previously unknown pathogens by initiating different defensive pathways. After successful defense and initiation of healing, the immune system returns to a state of homeostasis and assumes a wait-and-see role. All of this is achieved by complex mechanisms that are controlled and balanced by multiple activating and inhibitory feedback loops [35]. Thus, in an appropriate inflammatory response, there is a balance between adequate cytokine production to clear invaders, on the one hand, and avoidance of a hyperinflammatory response, in which an excess of mediators causes clinically significant collateral damage, on the other.

Cytokines play a pivotal role in these control mechanisms by regulating the immune response, which they can, thus, amplify but also dissolve. By default, their comparatively short biological half-lives prevent remote effects outside the inflammatory foci. In the case of disseminated infections, increased levels of circulating cytokines may also occur, although this is generally considered pathological [5]. However, it is precisely this systemic effect that can lead to collateral damage to various vital organ systems. Numerous pro- and anti-inflammatory factors are involved in the context of a dysregulated inflammatory response, as occurs in CS. In addition to cytokines and factors of the complement and coagulation systems, cellular responses—mediated by, e.g., monocytes, macrophages, neutrophils, NK cells, and endothelial cells—also play a role [35].

A dysregulated inflammatory response can have several causes: excessively high pathogen load in the context of sepsis, inadequate sensing or triggering of the immune system without the presence of a pathogen at all (as occurs with Castleman’s disease), or inappropriate inflammasome activation due to genetic disease [36,37,38]. Further examples are the inability of the immune system to terminate an initially adequate immune response and return to baseline (e.g., primary hemophagocytic lymphohistiocytosis, HLH) or conditions with uncontrolled infection and persistent immune activation (e.g., macrophage activation syndrome (MAS)-HLH in, e.g., CMV, EBV, or Influenza) [39]. Common to these syndromes are absence or failure of negative feedback control, which usually prevents hyperactivation of the inflammatory response. The excessive release of proinflammatory factors ultimately leads to systemic damage and even multi-organ failure.

## 3. Inflammation Due to Sepsis

Adaptive and innate immunities rely on a multitude of different soluble, intracellular, and membrane-bound receptors. Pattern recognition receptors (PRR) not only recognize pathogen-associated molecular markers (PAMPs, e.g., endo- and exotoxins, DNA, lipids) of foreign invaders, but also endogenous host-derived danger signals (damage-associated molecular patterns, DAMPs). 

The interaction of Toll-like receptors (TLRs) located on the membrane surfaces of antigen-presenting cells (APCs) and monocytes with PAMPs or DAMPs results in the initiation of signaling cascades and the expression of genes involved in inflammation, adaptive immunity, and cellular metabolism. This leads to the expression of so-called “early activation genes” and to the release of cytokines (e.g., IFN-γ, IL-1, IL-6, IL-8, IL-12) and components of the complement and coagulation systems. The complex pathways are explained in detail elsewhere [40,41,42].

This systemic increase of pro- and anti-inflammatory cytokines in the early phase is considered the classic hallmark of sepsis. The proinflammatory components cause inflammation which, if systemic, can lead to progressive tissue damage and to organ dysfunction. Concomitant immune suppression caused by downregulation of activating cell surface molecules increases apoptosis of immune cells, and depletion of T cells leads to “immune paralysis” in later stages of the disease course, making the organism susceptible to nosocomial infections, opportunistic pathogens, and viral reactivation [43,44].

Neutrophils are part of the first line of defense against microbes and, as a component of the innate immune system, may contribute to hyperinflammation in sepsis through the release of proteases and reactive oxygen species. Severe bacterial infections cause the release of both mature as well as immature forms of neutrophils after emergency granulocyte formation from the bone marrow. When activated by PAMPs or DAMPs, they show phagocytosis activity as well as oxidative burst capacity; additionally, they can release neutrophil extracellular traps (NETs) [45].

NETs are diffuse extracellular structures consisting of a network of chromatin fibers, antimicrobial peptides, and proteases such as myeloperoxidase, cathepsin G, and elastase. NETs contribute to antibacterial defenses because of their potential to trap and eliminate a wide range of pathogens, including Gram-positive and Gram-negative bacteria, viruses, yeasts, as well as protozoa and parasites that cannot be phagocytosed [46,47]. In animal studies, restriction of NET formation led to increased bacteremia and, thus, to a lower survival rate of test animals with sepsis [48]. However, excessive NETosis in sepsis can also be harmful. Large numbers of NETs in tissues or vessels due to excessive release or inadequate removal are associated with hypercoagulation and endothelial damage. NETs are rich in histones, and the binding of NETs to endo- or epithelia can lead to cell damage, both directly by NETs and histone modification. This can lead to the formation of intravascular thrombi and even multiple organ damage. Release of NETs has also been reported by cytokines (e.g., IL-8, IL-1β, TNF), platelet agonists, and antibodies [49,50,51].

A further essential component of innate immunity is the complement system. In the early phase of hyperinflammation, increased levels of activated complement factors such as the proinflammatory anaphylatoxins C3a, C4a, and C5a can be detected [52]. Anaphylatoxins, particularly C5a, have been shown to significantly contribute to response amplification, ranging from inducing apoptosis and paralyzing neutrophils up to boosting further instances of CS [53]. Increased C5a is associated with a worse clinical course due to increased systemic inflammation and apoptosis. C5a plays a role in neutrophil chemotaxis; by binding to the C5a receptor (C5aR), neutrophils gain the ability to migrate and enter inflamed tissue [54]. There, activation occurs through PAMPs and DAMPs with the release of granular enzymes, reactive oxygen species, and NETs.

During evolution, complement and coagulation systems have developed from a single pathway. The release of the strongly proinflammatory anaphylatoxins C3a and C5a in the context of complement activation also causes the recruitment and activation of platelets, endothelial cells, and leukocytes. The activation of the human contact system, or intrinsic coagulation in the form of coagulopathy, is presently also understood as part of the innate immune response [55]. Coagulation is activated by factor XI or cleavage of kininogen with release of bradykinin and antimicrobial peptides. In various experimental models, it could be shown that the inhibition of coagulation led to an impairment of the antimicrobial defense. In 2013, Engelmann and Massberg introduced the term “immunothrombosis” [56]. Indeed, highly preserved links between inflammation and hemostasis have been identified in mammals. Some coagulation factors are capable of inducing the release or activation of cytokines and, thus, potentially contribute to CS [57]. 

In sepsis as a specific form of hyperinflammation, coagulopathy is also a frequent complication, which can be detected in up to a third of critically ill patients and can lead to the development of multiple organ failure in severe cases. DIC is described by the International Society on Thrombosis and Haemostasis (ISTH) as a syndrome “*characterized by the intravascular activation of coagulation with loss of localization arising from different causes. It can originate from and cause damage to the microvasculature, which if sufficiently severe, can produce organ dysfunction*” [58]. The occurrence of DIC in sepsis represents a consumptive coagulopathy due to suppressed fibrinolysis with concomitant system-wide coagulation activation, which, in conjunction with systemic inflammation, can lead to organ dysfunction. For this, the term sepsis-induced coagulopathy (SIC) has been introduced, which is based on existing organ dysfunction, decreased platelets, and increased PT-INR [59].

The endothelium and its protective layer of glycoprotein polysaccharides (glycocalyx) play a critical role in disease progression during CS. The endothelium and glycocalyx are the sites of action of a variety of mechanisms that lead to an inflammatory response. Thus, endothelial cells, in turn, become drivers of coagulopathy: they lose anti-thrombotic properties, the expression of surface-bound thrombomodulin is reduced, and there is increased expression of tissue factor (TF), which in turn, together with leukocytic microparticles and monocytes that are also TF-occupied, leads to coagulation activation. Combined with the release of other proinflammatory factors of their own, there is increased recruitment of inflammatory cells, further expression of adhesion molecules, progressive hyperpermeability, and release of cytokines. Complex formation with coagulation factor VIIa (F VIIa) results in activation of the coagulation cascade via factors IX and X. Microbes as well as various cytokines and factors of the complement system cause the increased expression of TF on endothelial cells, macrophages, and monocytes [60].

An additional enhancement of the prothrombotic situation occurs through the binding of released TF to activated platelets and neutrophils, among others, while at the same time, the activity of the antithrombotic effect of antithrombin, the protein C system, and the tissue factor pathway inhibitor (TFPI) is reduced [61].

## 4. Unleashing the Cytokine Cascade

### 4.1. Sepsis

The most frequent cause of CS is invasive microbial infection. A proportion of infected patients develop a dysregulated immune response and the clinical appearance of sepsis as a life-threatening condition. At present, the Third International Consensus (Sepsis-3) emphasizes the crucial role of the innate and adaptive immune responses in the development of the clinical syndrome sepsis by defining it as “organ dysfunction caused by a dysregulated host response to infection” [62]. Sepsis affects approximately 49 million people annually. Estimates suggest that up to 11 million deaths occur annually due to sepsis, representing approximately 19.7% of all global deaths. Although global sepsis mortality rates appear to be decreasing, they are still as high as 25% in septic adults hospitalized in high-income countries [3]. In particularly severe courses with pronounced circulatory, cellular, and metabolic disturbances, known as septic shock, the hospital mortality rate may reach almost 60% [63].

It is difficult to distinguish between adequate cytokine production to fight systemic infection and dysregulated cytokine production. Disseminated microbial infections and the recognition of PAMPs induce the production and release of numerous cytokines, which subsequently leads to fever, blood pressure decrease, cell death, coagulopathy, and multiple organ dysfunction. The immune response can be a significantly greater threat to the host, through collateral damage to various tissues and organs, than the infection itself.

Various Gram-positive bacteria such as streptococci and staphylococci can produce so-called superantigens [64]. These bacterial superantigens are exceptionally potent mitogens; concentrations of less than 0.1 pg/mL are sufficient to lead to polyclonal T-cell activation by cross-linking T-cell receptors and the major histocompatibility complex (MHC) [65]. In addition, the presence of circulating lymphotoxin in patients with streptococcal toxic shock syndrome could be demonstrated by Sriskandan et al., thereby illustrating the specific activation of T cells [66]. Toxic shock syndrome may be the consequence, which is an immediate threat to the survival of the affected host.

Some patients with an exaggerated immunological response towards infection have defects in pathogen recognition, regulatory mechanisms, or mechanisms responsible for termination/resolution of the inflammatory response. For example, patients with a specific perforin disorder develop HLH-associated CS when infected with cytomegalovirus or Epstein–Barr virus [39]. Perforin usually participates in the termination of the inflammatory response, but the defective form appears to lead to impaired cytolysis which, in turn, prolongs the interaction between APC and lymphocytes and influences the clearance of antigen-bearing dendritic cells. This results in a self-sustaining loop of autocrine proinflammatory cytokine expression, continuous activation of macrophages and T cells, and sustained hemophagocytosis [67,68].

In summary, sepsis defined as life-threatening organ dysfunction due to a dysregulated host response to infection is the most common cause of CS. Elicited by PAMPs of bacterial origin and depending, to some extent, on pathogen load, the immediate systemic reaction consists of profound CS and immediate organ failure and shock.

### 4.2. Post-Cardiac Arrest Syndrome

Another example of the onset of CS is in the context of post-cardiac arrest syndrome (PCAS). In 2016, the American Health Association’s registry reported about 350,000 cases of out-of-hospital cardiac arrest (OHCA) and approximately 200,000 cases of in-hospital cardiac arrest (IHCA) in the USA [69]. The rate of return of spontaneous circulation (ROSC) was 45–50%, with high mortality before hospital discharge [70]. The development of PCAS with ischemia–reperfusion injury, hypoxic brain injury, and continued myocardial dysfunction seem to play an important role, in addition to the primary disease [71].

Cardiac arrest leads to global hypoxemia and organ damage due to no flow or low flow. After reperfusion, oxidative damage and the formation of free radicals lead to tissue damage, to the activation of different metabolic cascades, and to the release of proinflammatory cytokines [72]. Endothelial cells release TNF and IL-1β, which subsequently promotes further cytokines such as IL-6, IL-8, and IL-10 [73]. Systemic levels of IL-6 and IL-8 are associated with neurologic and cardiovascular impairment and mortality.

IL-6 and IL-8 concentrations increase both systemically and in the cerebrospinal fluid (CSF). IL-8 is known to have beneficial effects on neuronal growth and hippocampal neuronal survival, but similar to IL-6, it also causes increased blood–brain barrier (BBB) permeability, leading to an enlargement of ischemic areas, propagation of cerebral edema, and also to activation of the complement system [74]. This in turn leads to further systemic inflammation, endothelial activation, and the perpetuation of a persistent hemodynamically unstable state, giving rise to further damage to the heart and brain [75].

CS associated with PCAS also has a direct impact on cardiac performance. Increasing levels of IL-1β, IL-6, and TNF observed shortly after ROSC both decrease systemic vascular resistance and impair myocardial function [76]. TNF and IL-6 cause the expression of cell adhesion molecules (CAMs), promoting endothelial inflammation and further organ damage [77]. IL-6-mediated impairment of the endothelium and glycocalyx results in progressive vasodilation and capillary leakage with increasing circulatory instability [77,78]. Syndecan-1 and thrombomodulin are markers of endothelial damage and are associated with the severity of PCAS [75].

In summary, circulatory arrest and reperfusion injury are prime examples for systemic inflammation unrelated to infection. CS may occur in the most severe cases and will further intensify the initial damage.

### 4.3. Endotoxin

Lipopolysaccharides (LPS, endotoxin) are one of the most important virulence factors of Gram-negative bacteria and have an extraordinarily high pathogenicity to humans. LPS make up approximately 75% of the outer membrane of Gram-negative bacteria and are primarily responsible for the activation of innate immunity [79,80]. They are recognized by specific and highly conserved PRRs, stimulate the release of proinflammatory cytokines, and lead to an exuberant proinflammatory host response [81]. 

Chemically, LPS are glycolipid macromolecules consisting of an oligosaccharide core, an outer O-antigen polysaccharide, and a lipid A domain. Picomolar amounts of this lipid A are sufficient to activate macrophages and induce the expression of proinflammatory cytokines such as IL-1β and TNF [82,83]. Further, the direct injection of LPS into various tissues (brain, liver, heart, and others) has been shown to induce detectable levels of cytokines that may contribute to systemic cytokine levels [84,85,86]. In a case report of a person who, in an attempt to cure a tumor condition, administered a 3750-fold greater dose of LPS to herself compared to the standard dose for normal volunteers in experimental studies (4 ng/kg), full clinical manifestation of septic shock syndrome occurred [87].

The subsequent complex intracellular processes leading to the activation of proinflammatory cytokines and type-1 interferon genes are described in detail elsewhere [88,89,90,91]. In patients with meningococcal-associated CS, different circulating cytokines have shown to be highly correlated to plasma LPS levels, including TNF, IL-1β, IL-6, IL-8, and IL-10 as well as MCP-1 and MIP-1a [92,93,94,95].

In summary, lipopolysaccharides (LPS, endotoxin) are powerful activators of the human innate immune system and play a primary role in the early recognition of bacterial infections and in the stimulation of antibacterial defense. Typically, in Gram-negative infections, endotoxin is the one exogenous agent that unleashes CS in humans.

### 4.4. Post-Splenectomy Syndrome

Another example of a dysregulated immune response in the setting of CS is overwhelming post-splenectomy infection (OPSI). This may occur in splenectomized patients who have an increased susceptibility to infection and are at risk for increased mortality and morbidity. Common pathogens are encapsulated bacteria, of which *Streptococcus pneumoniae* infections are by far the most common cause of being associated with a mortality rate of up to 60% [96]. However, recent studies suggest that also *Haemophilus influenzae* (type b) and *Neisseria meningitidis* are other common pathogens as well as Pseudomonas species and *Escherichia coli* [97]. Although the definition is not standardized, OPSI can be considered a fulminant course of sepsis [98]. Non-specific symptoms such as fever, chills, and gastrointestinal symptoms may be followed by the full clinical picture of septic shock with DIC, hypoperfusion, and MOF within 48 h due to increased vascular permeability and profound vasoplegia caused by high levels of nitric oxide and prostaglandin.

The spleen plays an important role in choline-mediated hypo-inflammatory inflammatory control [99]. In sepsis, splenic macrophages are considered potent producers of TNF. Vagal interference results in a marked reduction in the expression of TNF and other proinflammatory cytokines, with concomitant increased release of anti-inflammatory cytokines such as IL-10. This anti-inflammatory influence is absent after splenectomy [100]. Further, clearance of attacking microbes is impaired because of delayed and impaired production of immunoglobulins and reduced phagocytic function [101]. Opsonization of microbes is also reduced, resulting in an increased risk of infection and a severe course of disease after splenectomy. Different animal models have demonstrated the importance of the spleen as a mediator in LPS-induced CS with regards to splenic macrophages or the activation of nonsplenic IL-6 production [102,103].

In summary, OPSI in splenectomized or asplenic patients may progress quickly from flu-like symptoms to CS, leading to fatal septic shock within 12 to 24 h [104].

### 4.5. Invasive Meningococcal Disease (IMD)

In about 10% of the healthy population, Gram-negative diplococcus *Neisseria meningitidis* can be detected on the mucosal surface of the nasopharynx as colonization [105]. Only a small proportion of the different strains present cause IMD. The disease begins as soon as meningococci enter the bloodstream and replicate. Systemic spread leads to different clinical forms of IMD with central or peripheral manifestations such as meningitis, purpura fulminans, pneumonia and, less commonly, septic arthritis or pericarditis [81,106].

Each year, approximately 500,000 people worldwide develop invasive meningococcal disease. Infants and young children are particularly affected because their immune system is not yet fully mature. A second age peak of disease is seen in adolescents [107,108]. Meningococcal disease of serogroup B, which affects up to 80,000 people worldwide annually, is associated with a high morbidity and mortality of up to 15%. In fulminant cases, death can occur in less than 4 h after infection [109,110]. Interestingly, patients with meningococcal disease have been the first cohort of patients in whom circulating cytokines (namely IL-1, IL-6, and TNF) have been detected for the very first time [111,112,113]. Numerous reports have shown a correlation between levels of proinflammatory immune response and outcome in IMD, especially for TNF and IL-1ß, but initially high levels of anti-inflammatory IL-10 accompanied by elevated proinflammatory cytokines are also associated with increased mortality [114,115,116].

Different strains of meningococci have unique virulence factors that promote adhesion, colonization, and invasion of the bloodstream. In serogroup B meningococci, for example, a similarity of the polysaccharide to polysialic acid structures of the human cell adhesion molecule ensures only low immunogenicity. Sialylated lipooligosaccharide (LOS) and factor H-binding protein (fHBP) support bacterial survival in the bloodstream, as they help the microbes to resist the host’s proinflammatory response, e.g., antibody recognition and phagocytosis by innate immune cells [117]. Thus, meningococcal factor H-binding protein circumvents the bactericidal effect of the complement system by inactivating host complement fHBP [106,118].

A critical factor in the pathogenicity of meningococci is the ability to interact with endothelial cells, including those of the BBB [106]. This causes endothelial dysfunction such as vascular leakage, microthrombi, and necrotic purpura, providing a niche in which the pathogen can replicate, leading to continuous bacteremia and high mortality [119].

On the cell membrane of meningococci, type IV pili (T4P) are localized, which are widely distributed in bacteria and are physiologically involved in different regulatory mechanisms. Known modes of action of this so-called *“prokaryotic Swiss Army knife”* include adhesion to abiotic and biotic surfaces, biofilm formation, motility, aggregation, and DNA uptake [120,121].

When colonizing human endothelia, meningococcal T4P first interact with CD147, a surface protein of the immunoglobulin superfamily that is expressed on different cell types and consists of two Ig-like domains extracellularly [122]. CD147 can strongly stimulate the secretion of proinflammatory cytokines as well as further activation of B and T cells through downstream-mediated activity of the JAK/STAT pathway [123,124].

In summary, invasive meningococcal disease is characterized by a fulminant response of the innate immune system. Imbalanced systemic inflammation (CS), coagulopathy, and microvascular injury are driving the pathophysiology. IMD is recognized as a prototypical endotoxin-driven Gram-negative disease involving virtually all inflammatory cytokines and mediators.

### 4.6. Viral and Parasitic Infections

“Sepsis should be defined as life-threatening organ dysfunction caused by a dysregulated host response to infection” [62]. The Sepsis-3 definition can be effortlessly applied to the severe forms of viral as well as parasitic infections since it emphasizes the role of the host response towards an invading pathogen.

Infection with certain viral pathogens is frequently accompanied by an excessive release of cytokines. These include MERS-CoV, influenza virus (e.g., H1N1, H5N1), and hemorrhagic fevers (e.g., Dengue, Ebola, and Crimean-Congo virus infections) [125,126,127,128,129]. Different viruses are associated with diverse patterns of cytokine release, but so far, this has not been operationalized in terms of a comprehensive and universally effective therapeutic approach. Encompassing both the pathogen and the host response, functional genomics seems to have the potential to provide a deeper understanding of infectious diseases. Modern molecular biology technologies such as microarrays provide a global view of changes in gene expression triggered by a variety of stimuli and allow for simultaneous profiling of thousands of transcriptional variations in an organ or tissue compartment. In 2012, Tisoncic and colleagues published findings on CS in the context of various infections, with a special focus on respiratory viruses using next-generation sequencing [6].

Parasitic infections are also associated with the development of CS; there are corresponding reports for infection with plasmodium falciparum and visceral leishmaniasis, among others. In research on Malawian children with different manifestations of malaria, it could be shown that the simultaneous occurrence of high levels of pro- and anti-inflammatory cytokines could contribute to the pathogenesis of cerebral malaria [130].

Visceral leishmaniasis is a protozoan infection caused by leishmania infantum chagasi. The systemic disease is based on a complex host–parasite interaction, where the parasite primarily affects cells of the macrophage lineage, resulting in marked T-cell depletion and reduction of hematopoietic cells. Damage to gut-associated lymphoid tissue allows bacteria to enter the bloodstream and stimulate macrophages through LPS. As a result, proinflammatory cytokines and other soluble factors such as migration inhibitory factor (MIF) are released, which in turn activates lymphocytes. Sustained and excessive stimulation ultimately leads to exhaustion of the T-cell compartment and, subsequently, to immunosuppression [131].

The term CS has re-emerged with the occurrence of the COVID-19 pandemic. Although not all mechanisms of SARS-CoV-2 virus-induced lung injury have been fully elucidated to date, CS has almost become synonymous with it, both in the scientific community as well as in the mass media. Therapeutic agents that interfere with cytokines, such as the monoclonal antibody tocilizumab or the Janus kinase (JAK) inhibitors baricitinib and tofacitinib, are used in the treatment of COVID-19. In each case, the justification is a need to control the dysregulated host response. Furthermore, there is ample evidence that ubiquitous viral spread affects multiple organ systems [132,133].

In many cases, pronounced hyperinflammation has been observed in severe courses with often lethal outcome, which resembles the course of other hyperinflammatory syndromes. Especially in patients with an unfavorable clinical course, the typical appearance is an exuberant immune response with hyperactivity of dendritic cells, lymphocytes, macrophages, and other immune cells, leading to a self-sustaining and self-amplifying pathophysiology. Usually, elevated levels of inflammatory markers such as IL-6, IL-8, IL-10, ferritin, and CRP are found in these patients; however, CS, sensu stricto, can only be detected in a minority of cases, and some authors claim that “the storm is rather a breeze” [134,135,136,137]. Yet, primary major compartmentalized CS in the context of COVID-19 occurs regularly in the lungs [138,139].

Published trials with severe courses of COVID-19 reporting IL-6 levels were analyzed in a meta-analysis by Leisman et al., and the results were compared with studies including patients with COVID-19-independent conditions such as ARDS, sepsis, and CRS [140]. In summary, they showed that systemic levels of IL-6 in patients with COVID-19 were about 12 times lower than in ARDS, about 27 times lower than in patients with sepsis, and as much as 100 times lower than in CRS.

Webb et al. have developed a COVID-19-specific hyperinflammation score (cHIS) that uses six parameters to assess the inflammatory situation in COVID-19 [141]: fever, macrophage activation (hyperferritinemia), hematological dysfunction (neutrophil-to-lymphocyte ratio), hepatic injury (lactate dehydrogenase or aspartate aminotransferase), coagulopathy (D-dimer), and cytokinemia (C-reactive protein, interleukin-6, or triglycerides). Applied to a cohort of nearly 300 COVID-19 patients, the score was associated with a 95% sensitivity and 59% specificity predicting the need for mechanical ventilation and 96% sensitivity/49% specificity predicting mortality, making hyperinflammation a potential major factor for poor clinical outcome in COVID-19 patients, in addition to functional immunoparalysis, coagulopathy, and direct viral cell injury.

In summary, among viral and parasitic infections, influenza, hemorrhagic fevers (Dengue, Ebola), malaria, and visceral leishmaniosis are most likely to cause CS. In SARS-CoV-2, however, CS does not seem to play a major role other than what has been previously reported at the beginning of the pandemic.

### 4.7. Sterile Inflammation and Iatrogenic CS

There are various publications and reports about the clinical picture of sterile inflammation. One of the best documented examples is the case of six healthy volunteers who participated in the first phase I study of TGN1412, a novel anti-CD28 monoclonal antibody and superagonist, in 2006 [142]. Within 90 min of receiving a single intravenous dose of the drug, all six volunteers showed signs of a progressive systemic inflammatory response, characterized by a rapid induction of proinflammatory cytokines as well as unspecific clinical symptoms (e.g., nausea, diarrhea, erythema, vasodilatation, hypotension, and myalgias). Within 12 to 16 h, the clinical situation deteriorated and pulmonary infiltrates, renal failures, and disseminated intravascular coagulation developed. Additionally, severe depletion of lymphocytes and monocytes occurred within 24 h after infusion. All six patients were admitted to intensive care units (ICU), requiring cardiopulmonary support for many weeks. In the absence of contaminating pathogens and endotoxins, this clinical course impressively demonstrated a dysregulated inflammatory response. Similar courses, albeit less dramatic, were also observed for anti-CD3 and anti-CD20 antibodies [143,144].

As another salient example for sterile (or iatrogenic induced) inflammations, the therapeutic use of CAR-T cells should be mentioned; as a modality of immunotherapy, CAR-T cell therapy is used for certain forms of acute lymphocytic leukemia (ALL) as well as some types of non-Hodgkin lymphoma (NHL) [145,146]. The patient’s own T cells are extracted and equipped with chimeric antigen receptor ex vivo by the use of viral vectors. This allows the genome information of the receptors to be maintained even during activation and division of these T cells. The receptors encoded by this consist of an extracellular binding domain, a linker region, as well as a transmembrane domain and an intracellular signal sequence [147]. Tumor cells that were previously “invisible” to the immune system are recognized by the binding domain through a specific antigen structure, which leads to adhesion and activation of the CAR-T cells.

These highly activated CAR-T cells are directed against the surface protein CD19, which is found on almost all lymphoma cells, but also on natural B lymphocytes. Reimplantation is the trigger of CS, with high systemic levels of IFN-γ and IL-6 already after a few hours to days after reinfusion [148]. These cytokines lead to the activation of further immune cells followed by typical symptoms such as fever, headache, drop in blood pressure, and neurological symptoms up to life-threatening CS with corresponding damage to organ systems. With organ support and anti-inflammatory measures, CS often regresses after a few days and affected organ systems recover.

In addition to the specific manifestations described above, an entity of sterile inflammation is observed much more frequently in everyday clinical practice: CS in the context of an acute, primarily non-infectious pancreatitis. In these cases, proinflammatory and anti-inflammatory responses oppose each other, whereby an imbalance between these two systems leads to localized tissue damage as well as distant organ dysfunction [149].

In summary, CAR-T cell therapy is the most prominent example for iatrogenic CS. However, for other targeted interventions in oncology and transplantation medicine such as rituximab, muromonab (OKT3), blinatumomab, and alemtuzumab, cytokine release syndrome (CRS) or CS has been frequently reported [150].

## 5. After the Storm

Counterintuitively, at first glance, the elimination of pathogens causing septic shock does not guarantee complete convalescence. Thanks to advances in intensive care medicine, many patients survive CS and shock. However, after many host-damaging events, the image of scorched earth remains. Even though acute care medicine traditionally focuses on the early phase of sepsis, essential and outcome-determining immunological changes (i.e., leukocyte reprogramming, endotoxin tolerance, and acquired immunoparalysis) occur later (Figure 1).

In early phases of the course of sepsis, low B- and T-lymphocyte counts are often found [152]. The cause of this septic lymphopenia, which is associated with increased mortality if it persists, has not been extensively elucidated, but is apparently based on a variety of mechanisms. In addition to increased apoptosis and tissue migration, there is also decreased production of lymphocytes as a result of emergency hematopoiesis, which prioritizes the production of monocytes and neutrophils and flushes immature myeloid-derived suppressor cells (MDSCs) into the peripheral blood [153,154]. There, they become functionally active and release anti-inflammatory cytokines (e.g., IL-10 and transforming growth factor β, TGF-β), resulting in marked immunosuppression. Recently, using single-cell RNA sequencing, different MDSC subsets could be detected, each of which could be used as a prognostic factor for the different courses of sepsis-related diseases. This makes MDSC a worthwhile approach in future research of septic mechanisms [155]. Other causative factors of lymphopenia include increased migration into tissues and increased apoptosis. 

Another component of immune paralysis is a markedly decreased expression of HLA-DR on the surfaces of monocytes and dendritic cells, which impairs pathogen recognition by decreased opsonization and impedes the Th1 and Th2 response as an essential part of the adaptive immune response (“immunological synapse”) [156]. If monocytes fail to restore HLA-DR expression, the clinical outcome is likely to be unfavorable [157,158]. 

In addition to the reduction in lymphocytes, increased apoptosis of monocytes and APC occur in a clustered fashion during the later course, which significantly reduces the production of proinflammatory cytokines [159]. Controlled apoptosis of innate and adaptive immune cells may initially be beneficial. However, uncontrolled reduction of the inflammatory response leads to incompetence in defense against further invasive microbes. Attempts to suppress immune cell apoptosis in sepsis have been shown to be promising [160]. 

A significant proportion of sepsis survivors develop persistent critical illness (PCI) with ongoing organ dysfunction and markedly impaired quality of life [161]. A proportion of these PCI patients also develop a clinical syndrome of persistent inflammation, immunosuppression, and catabolism (PICS), as described in 2012 by Gentile et al. for patients who had an ICU stay >10 days with a surgical diagnosis [162]. Initially described as “compensatory anti-inflammatory response syndrome” (CARS), “late MOF”, or “complicated clinical course”, typical PICS develops after major trauma, major surgery, or pronounced inflammatory or septic insult. It is characterized by persistent inflammation with acquired immunosuppression, resulting in a prolonged ICU stay and, ultimately, a poor prognosis [163].

## 6. Viral- vs. Non-Viral-Induced CS

By and large, the clinical picture of infective systemic inflammation is uniform, regardless of whether there is an underlying bacterial or viral cause. Even non-specific, non-infectious inflammation such as trauma or allergic reactions present with symptoms such as fever, increased respiratory rate, and tachycardia. Recognizing the true origin of an inflammatory host response often presents a significant challenge for treating physicians, not least in applying the right therapy.

Based on current knowledge, the endogenous processes of host defense in bacterial and viral infections differ in several aspects. The immune response to infection by bacteria is initially fundamentally different from that to viruses: In simple terms, the innate immune system recognizes and destroys bacteria, which are mostly extracellular, with the help of the classical and alternative complement pathway and phagocytosis, among other mechanisms. Viruses, on the other hand, replicate intracellularly and infected cells are recognized and destroyed largely by the adaptive immune system. In this process, cytotoxic T cells, which recognize viral epitopes presented by MHC-I molecules, antibodies, and interferons play key roles. The development of CS from the various underlying basic mechanisms is a multifactorial process with numerous variables, some of which will be discussed here as examples.

In addition to SARS-Cov-2, influenza infection is also frequently associated with the initiation of CS and both are single-stranded RNA viruses. Influenza requires viral RNA polymerase to synthesize an mRNA from the viral genome for replication [164]. The presence of viral RNA in the host cytoplasm activates three different immune pathways: Toll-like receptors (TLRs, mainly TLR3 and TLR7), retinoic acid-induced gene-1 protein (RIG-I), and Nod-like receptors (NLR). This initiates the innate immune response to influenza virus, which, among other things, initiates the production of type I and III interferons and also activates the nuclear factor kappa-light-chain-enhancer (NF-κB) of activated B cells [165,166]. Viral RNA also activates inflammasomes, thereby releasing IL-1β and IL-18 [167]. If the inflammatory reaction is very pronounced, an ultimately uncontrolled release of proinflammatory cytokines leads to cytokine storm with the known potentially lethal courses [168].

Interferon (IFN) type I and III are archetypal cytokines that play a major and well-established role in antiviral immunity. After recognition of viral patterns by basic PAMP and PRR mechanisms, their release induces hundreds of interferon-stimulated genes (ISGs) [169,170].

Recent evidence suggests a key role of IFN also in bacterial infections in terms of enhancing anti-bacterial host responses [171,172]. Thus, IFNs are an attractive virulence target for bacterial pathogens. Bacteria secrete effectors that inhibit IFN type I and III production and signaling, thereby also affecting ISG expression and function [173]. Characterizing shared pathways of viral and bacterial inflammation is of great interest for the development of effective therapeutic approaches.

## 7. Age-Related Changes of the Immune System

In developed countries, demographic changes result in increasingly elderly patients being hospitalized with sepsis, who, due to immunosenescence, may respond less effectively to pathogen- or damage-associated molecular patterns [174].

Although the increased incidence and mortality in the elderly in infectious diseases has been associated with immunosenescence, the links between immunosenescence and sepsis have been poorly studied [175,176]. With increasing age, the human immune system changes: the effectiveness of both the adaptive and innate immune responses decreases. Tissue and cellular repair capacities are limited with increasing age, and the ability to build up a vaccine response also decreases. These processes are referred to as “immunosenescence” and describe the age-related dysfunction of both innate and adaptive immunities [177,178].

Interestingly, immunosenescence leads to an overall increase in the activity of the innate immune system. Proinflammatory cytokines (e.g., IL-6, TNF, CRP) are released more frequently and the basic composition of the cells of the defense system also changes [179]. These processes collectively lead to a low-grade and chronic inflammatory response, termed “inflammaging,” which is associated with the biological aging process. Various age-related diseases such as Alzheimer disease, hypertension, arthritis, atherosclerosis, but also type 2 diabetes and cancer are attributed to inflammaging [180,181].

In sepsis, immunosenescence plays an important role in increasing older patients’ susceptibility to sepsis with poor prognosis [182,183]. Overall, the diagnosis of sepsis in older adults is sometimes significantly more difficult. Older patients often present with non-specific or atypical symptoms of infection, such as altered mental status, weakness, dizziness, loss of appetite, and general malaise [184,185,186]. Initial signs of a systemic inflammatory response, such as fever, may even be absent [187].

One of the key immune alterations in sepsis is delayed neutrophil apoptosis, which leads to persistent neutrophil dysfunction when accompanied by an increase in immature neutrophils [188,189]. Although the overall neutrophil count is well-preserved in the elderly, chemotaxis, phagocytosis, and NET-formation are impaired [190]. In addition, decreased neutrophil migration accuracy combined with decreased efferocytosis and blunted responses were shown to result in prolonged immunosuppression, which partially explains increased mortality in elderly patients [191].

The other striking feature of sepsis-induced immunosuppression is the reduced ability of macrophages and monocytes, known as endotoxin tolerance, to respond to subsequent challenges with LPS or other inflammatory stimuli [192]. Although macrophages exhibit significant age-related functional changes, their numbers remain constant. However, in the elderly, the initial macrophage response to microbes and other inflammatory stimuli is reduced, likely due to age-related decreased TLR expression and associated downstream signaling [193]. Both aging and sepsis decrease HLA-DR expression, antigen presentation, and phagocytosis in macrophages and monocytes. 

The number of NK cells is mostly slightly increased in the elderly [194,195]. This is due to a slight increase in the more mature CD56^dim^ subset, with the number of immature CD56^bright^ subsets decreasing in the elderly [196]. Functionally, NK cells exhibit normal or increased IFN-γ production but decreased cytotoxicity, most likely explained by decreased release of perforins [197,198]. It is possible that the increased production of IFN-γ by NK cells with age, which in turn leads to increased tissue damage, is a cause of the more frequent occurrence in elderly patients [179].

Accumulating evidence points towards an important role of myeloid-derived suppressor cells (MDSC), members of a heterogeneous immature myeloid cell population, in immunosenescence. The number of MDSCs increases with age [199]. They are associated with sustained sepsis-induced immunosuppression by promoting T- and B-cell depletion, functional inhibition of DC and macrophages, and by inhibition of NK cells while promoting Treg [192,200].

## 8. Endotypes, GWAS, and Transcriptome Analysis

The syndromic nature of sepsis and the tremendous heterogeneity of affected individuals in terms of predisposition (age, sex, genetics, and comorbidities, to name only a few) and host response (e.g., distinct endotypes) most probably preclude the existence of a one-size-fits-all target or therapy. Emerging data from genome-wide association studies (GWAS) and transcriptome analysis have revealed distinct host response patterns within the classical phenotypes that might become future targets for individualized therapeutic interventions [201].

Further, designing clinical trials to identify patient populations with high propensity for benefit of the tested intervention by predictive enrichment could be a successful strategy. For example, low expression of human leukocyte antigen-DR on monocytes (mHLA-DR) has been used to increase the homogeneity of the patient population while enriching the patient population with a potentially beneficial response to therapy [202,203,204].

Significant advances in genomics, proteomics, and metabolomics (also referred to as the “-OMICS” approach) as well as in point-of-care diagnostics are enabling new therapeutic approaches [HASIN 2017]. However, the use of such gene expression assays is mostly still laborious and their use as point-of-care platforms is, therefore, limited. 

In a Dutch project, the exploration of patterns of different mRNA transcripts using machine learning techniques has led to the identification of four distinct sepsis endotypes. These subclusters differ in their immune states and have been named Molecular Diagnosis and Risk Stratification of Sepsis (MARS) 1 to 4. Cluster 1, for example, presents with a marked decrease in the expression of genes corresponding to important cellular functions of the innate and adaptive immune systems and a lower 28-day survival rate [205]. The UK Genomic Advances in Sepsis (GAinS) study also identified distinct subgroups of patients. The clusters, referred to here as SRS 1 and SRS 2, differ based on their immune status, with patients with SRS 1 showing immunosuppression and increased 14-day mortality [201]. Identifying genetic expression signatures as early as possible at first clinical presentation may be the key to identifying the most vulnerable patient groups early, to predict evolving severity, and to initiate appropriate therapy. Kreitmann and colleagues performed a multi-cohort analysis using pooled gene expression data from 1437 arrays sampled on day 1 following admission on sepsis patients, which provided relevant information for predicting 30-day mortality [206].

In 2015, Wong et al. published a study that used multiplex mRNA quantification to endotype the expression of the one hundred subclass-defining genes in children with septic shock [207]. Subclass A was associated with harmful effects of glucocorticosteroids due to a genetic mutation of the glucocorticoid receptor. Hence, at least in this context, identification of subclasses in real time has theranostic implications.

## 9. Future Perspectives

CS is a rare but catastrophic event occurring mostly in sepsis. Pathogen load, host factors, and pathogen–host interaction are crucial sine qua non elements.

While new insights have improved our understanding of the complex immunological processes, millions of people worldwide continue to die both “from” and “with” sepsis, and after three decades of research, effective therapies remain scarce. To address this vexing conundrum, important topics should be brought into the focus of research in the future, and past errors should be rigorously analyzed:(a)Preclinical models for both basic and translational research need to be improved; current animal models do not adequately represent inflammation. New models are needed to comprehensively represent the complex processes and numerous feedback loops and redundant pathways should be considered. The individual reactions of the components of the immune system within different compartments need to be further elucidated. Local microenvironments with different molecular and cellular characteristics can result in different polarizations of the immune response in different tissues and organs as mentioned above. It has been shown that the ex vivo phenotype of leukocytes is strongly dependent on their compartment of origin. Blood leukocytes may show limited proliferation and secretion capacity, with reduced expression of HLA-DR mRNA in monocytes, while recruited monocytes in the lungs express greater than threefold more HLA-DR on their membrane [208,209]. Further, lung, spleen, and peritoneal macrophages have disparate transcriptomes and cell surface marker expression [210];(b)Development of a personalized approach combining clinical phenotypes, endotypes, and biomarkers, deciphered through the combination of “-omics” technologies and artificial intelligence (AI), is needed.

AI-assisted phenotype analysis of large populations has resulted in the recognition of subclasses based on routine clinical and laboratory data. Seymour et al. identified four distinct clinical phenotypes that correlated with host response patterns and outcome parameters, and computer simulations suggested that these phenotypes could contribute to understanding heterogeneity in treatment effects. Along the same line, another study analyzing seven hundred transcriptomic profiles suggested that patients with bacterial sepsis could be divided into “inflammopathic”, “adaptive”, and “coagulopathic” clusters. For each of these clusters, taking different therapeutic approaches would potentially be superior to the one-size-fits-all paradigm [211]. 

Additionally, targeting specific pathways may potentially be beneficial in critically ill patients if that pathway is under- or overexpressed. Post-hoc analyses of clinical trials showed, for example, a benefit for the use of anti-TNF therapies, IL-1 receptor antagonists, or corticosteroids in patients with elevated ferritin and IL-6 levels or low IFN-γ/IL10 ratios [212,213,214]. In contrast, post-hoc analysis of another study found that a transcriptomic signature indicative of a more immunocompetent profile performed significantly worse under steroid therapy [215].

(c)Further, there is a significant need for diagnostic testing procedures that can quickly and reliably help differentiate, for example, between bacterial and viral bases and, thus, possibly guide the use of anti-infectives and ultimately counteract the development of resistance. The current routine involves culture, isolation, and identification of pathogens from patient material—a time-consuming and not necessarily successful process. Contamination, mixed cultures, and unsuccessful cultivation are common risks. In addition, depending on the (presumed) focus, obtaining appropriate samples is not always without risk (intrapulmonary, cerebrospinal fluid). Although detection and cultivation conditions are constantly being developed, the method of direct detection, e.g., by PCR or immunoassays, also has limitations—previously unknown variants may evade detection and the method cannot reliably distinguish between already dysfunctional/dead pathogens and inflammatory-active material [216,217]. Further research and an increased understanding of the interrelationships between what has been considered viral and bacterial defense cascades may lead to further blurring of the boundaries and to therapeutic options in which the exact origin of inflammation may turn out to be less relevant [218,219]. 

For these reasons, increasing attention is being directed towards the identification of host biomarkers that reflect different host responses to different triggers (viral, non-viral, or non-infectious) [220,221,222]. In 2017, Sampson et al. published a study in which they validated—based on two of their own studies as well as 44 publicly available datasets—a four-gene expression signature from whole blood that can be used to detect a specific host response to the presence to different virus-related infections [223]. Using the gene signatures of interferon-stimulated gene 15 (ISG15), interleukin 16 (IL16), adhesion G protein-coupled receptor E5 (ADGRE5), and 2′,5′-oligoadenylate synthetase-like (OASL) genes, it was possible to discriminate between viral and non-viral geneses using numerous human and non-human validation datasets. Because this viral signature relies on only four biomarkers, it could potentially become clinically useful when implemented on a point-of-care platform. Recently, Xu and colleagues presented a two-transcript biomarker with the RT-PCR transcripts of the host genes IFI44L and PI3 as a robust classifier to discriminate bacterial infections from viral infections in adults with acute febrile illness [224].

## 10. Summary

The immune system protects the host organism from exogenous and endogenous pathogens. A finely tuned and balanced array of cytokines, coagulation factors, and complement together with immunocompetent cells protect the body from a wide variety of known and unknown invaders. Usually, pro- and anti-inflammation are tightly regulated to adequately counter the infectious event. A proinflammatory milieu typically dominates the initial phase; however, anti-inflammation is initiated early to reach a new equilibrium and to start tissue repair processes.

Various pathogens and malignant and autoimmune diseases as well as genetic changes, but also iatrogenic interventions, can disturb this equilibrium so that an excessive release of cytokines can occur. In its severe (albeit rare) form, this is referred to as cytokine storm. As a result, positive feedback mechanisms and self-sustained activation of immune cells occur. The resulting hyperinflammation can lead to a life-threatening condition. Endotheliopathy, disseminated intravascular coagulation, microcirculatory disturbances, and profound hemodynamic alterations occur. Consequently, distal organ damage may develop, culminating in multiple organ failure and death in particularly severe cases. The damage caused by the immune response may be more severe than the damage caused by the pathogen itself. If patients survive, chronic critical illness with high morbidity and considerable long-term mortality can evolve. To date, the available therapeutic approaches are more or less limited to guideline-based sepsis therapy and symptomatic measures in the context of intensive care organ support. By analyzing the pathophysiology and through data-based pattern recognition, advances are conceivable in the future that will enable individualized and effective treatment.

Take-home messages:We support the comprehensive and clinical definition of CS that was recently suggested by Fajgenbaum and June [5]:(i)Elevated circulating cytokine levels;(ii)Acute systemic inflammatory symptoms;(iii)Severe secondary organ dysfunction.In CS, both excessive hyperinflammation and uncontrolled anti-inflammation occur simultaneously. Survivors of the initial phase often develop acquired immunosuppression, which is an additional risk factor for unfavorable outcomes and long-term morbidity;A wide array of infectious and non-infectious disease may cause CS; the most common cause is sepsis due to invasive microbial infection;The biomarker signature profiles of different types of CS are rather distinct. However, to use these signatures to diagnose the origin of CS is currently not yet feasible;CS characteristics differ between different compartments (blood, CSF, pleural effusion, lung tissue, etc.) In clinical routine, the blood compartment is almost exclusively used for analysis; this may lead to wrong conclusions and misconceptions of the underlying pathophysiology;The earliest possible detection of CS is of outstanding importance, as this may be related to therapeutic decisions and, ultimately, prognosis and outcome;Transcriptome analysis and GWAS represent promising opportunities for future development. These techniques enable improved knowledge of different phenotypes and may help to implement “precision medicine”, very similar to what is already done in modern oncology.

## Figures and Tables

**Figure 1 ijms-23-11740-f001:**
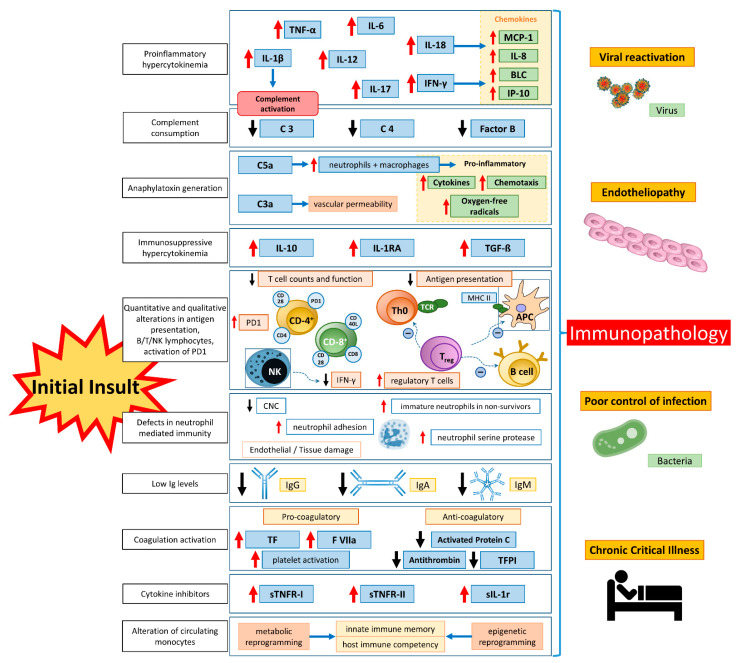
Aspects of immunological dysfunction caused by sepsis with details of the entities involved. APC, antigen-presenting cell; BLC, B-lymphocyte chemoattractant; CD, cluster of differentiation; CNC, critical neutrophil concentration; IFN-y, interferon y; Ig, immunoglobulin; IL, interleukin; IP-10, IFN-gamma-inducible protein 10; MCP-1, monocyte chemoattractant protein-1; MHC II, major histocompatibility complex II; PD1, programmed death protein 1; sIL-1r, soluble interleukin-1 receptor; sTNFR, soluble tumor necrosis factor receptor; TCR, T-cell receptor; TF, tissue factor; TFPI, tissue factor pathway inhibitor; TGF-β, transforming growth factor β. Adapted from Bermejo-Martin JF with permission [151].

**Table 1 ijms-23-11740-t001:** Common biomarkers affected during cytokine storm. BLC, B-lymphocyte chemoattractant; CCL, chemokine ligand; CRP, C-reactive protein; CXCL, CXC motif chemokine ligand; IL, interleukin; IP-10, IFN-gamma-inducible protein 10; MCP-1, monocyte chemoattractant protein-1; MIG, monokine induced by IFN-gamma; MIP-1, macrophage inflammatory protein 1; NK cell, natural killer cell; Teff cell, effector T cell; TH cell, T helper cell; Treg cell, regulatory T cell.

Mediator (Abbreviation)		Main Source	Major Function
**Cytokines**			
IL-1		macrophages, pyroptotic cells, epithelial cells	Proinflammatory; pyrogenic function; activation of macrophage and T_H_17 cells
IL-2		T cells	Immune response; T_eff_ and T_reg_ cell growth factor; T-cell differentiation
IL-4		T_H_2 cells, basophils, eosinophils, mast cells, NK cells	Anti-inflammatory; T_H_2 differentiation; adhesion; chemotaxis
IL-6		T cells, macrophages, endothelial cells	Proinflammatory; pleiotropic; pyrogenic function; acute phase response; lymphoid differentiation; increased antibody production,
IL-9		T_H_9 cells	Pleiotropic; stimulation of B, T, and NK cells; protection from helminth infections; activation of mast cells; association with type I interferon in COVID-19
IL-10		regulatory T cells, T_H_9 cells	Anti-inflammatory; inhibition of macrophage activation; inhibition of T_H_1 cells and cytokine release
IL-12		dendritic cells, macrophages	Stimulation of T and NK cells; activation of T_H_1 pathway; induction of interferon-γ from T_H_1 cells; cytotoxic T cells and NK cells; acting in synergy with interleukin-18
IL-13		T_H_2 cells	Anti-inflammatory; differentiation of B cells; mediator of humoral immunity
IL-17		T_H_17 cells, NK cells, group 3 innate lymphoid cells	Protection from bacterial and fungal infections; promotion of neutrophilic inflammation
IL-18		monocytes, macrophages, dendritic cells	Proinflammatory; activation of T_H_1 pathway; synergistic with interleukin-12
IL-31		T_H_2 cells, macrophages, mast cells, dendritic cells	Proinflammatory; cell-mediated immunity
IL-33		macrophages, dendritic cells, mast cells, epithelial cells	Proinflammatory; amplification of T_H_1 and T_H_2 cells; activation of cytotoxic T cells, NK cells, and mast cells
Type I Interferon		virtually all body cells	Dendritic cell activation/maturation/migration/survival; enhancement of the activity of NK and T/B cells; induction of antiviral effector molecules; antagonism to the action of interferon-γ
Interferon-γ(Type II IFN)		T_H_1 cells, cytotoxic T cells, group 1 innate lymphoid cells, NK cells	Proinflammatory; activation of monocytes and macrophages
Lymphotoxin α		activated lymphocytes	Pleiotropic; activation of NF-κB pathway
TGF-β		T_reg_ cells, monocytes, macrophages, fibroblasts, epithelial cells, cancer cells	Immunosuppressive; regulation of proliferation, differentiation, apoptosis, and adhesion; inhibition of hematopoiesis
Tumor necrosis factor		T cells, NK cells, mast cells, macrophages	Pyrogenic; increasing vascular permeability
**Chemokines**			
MCP-1	CCL2	macrophages, dendritic cells, cardiac myocytes	Pyrogenic; recruitment of T_H_1 cells, NK cells, macrophages, eosinophils, and dendritic cells
MIP-1α	CCL3	monocytes, neutrophils, dendritic cells, NK cells, mast cells	Recruitment of T_H_1 cells, NK cells, macrophages, and dendritic cells
MIP-1β	CCL4	macrophages, neutrophils, endothelium	Recruitment of B cells, CD4^+^ T cells, and dendritic cells
IL-8	CXCL8	macrophages, epithelial cells	Recruitment of neutrophils
MIG	CXCL9	monocytes, endothelial cells, keratinocytes	Interferon-inducible chemokine; recruitment of T_H_1 cells, NK cells, and plasmacytoid dendritic cells
IP-10	CXCL10	monocytes, endothelial cells, keratinocytes	Interferon-inducible chemokine; recruitment of T_H_1 cells, NK cells, and macrophages
BLC	CXCL13	B cells, follicular dendritic cells	Recruitment of T_H_1 cells, monocytes, dendritic cells, and basophils
**Plasma proteins**			
CRP		hepatocytes	Interleukin-6 increases CRP expression, interleukin-8 and MCP-1 secretion;
Complement		hepatocytes, other cells	In cytokine storm, activation of complement contributes to tissue damage, inhibition may reduce immunopathologic effects

**Table 2 ijms-23-11740-t002:** Diagnostic and prognostic biomarkers of sepsis. Angpt2/1, Angiopoietin 2/Angiopoietin 1; APACHE, Acute Physiology And Chronic Health Evaluation; AUC, Area under the curve; Bio-ADM, Bioactive Adrenomedullin; CaPT, Calprotectin; CRP, C-reactive protein; HMGB-1, High-mobility group protein B1; hs-CRP, high-sensitivity C-reactive protein; IL-6, Interleukin-6; MR-proADM, Mid-regional proAdrenomedullin; PCT, procalcitonin; PTX-3, Pentraxin-3; sCD14-ST, soluble CD14 subtype; SIRS, systemic inflammatory response syndrome; SOFA, Sepsis-related organ failure assessment; sTREM-1, soluble triggering receptor expressed on myeloid cells-1; suPAR soluble urokinase-type plasminogen activator receptor.

Study	Number of Patients	Diagnosis/Prediction	Commonly Used Markers/Comparators	New Biomarkers	Variables	AUC
Kweon et al. [11]	n = 118 (73 sepsis; 45 SIRS or healthy controls)	Sepsis	PCT IL-6	sCD14-ST (Presepsin)	sCD14-ST PCT IL-6 hs-CRP	0.937 0.915 0.869 0.853
Lu et al. [12]	115 patients (72 sepsis; 43 SIRS or healthy controls)	Sepsis	PCT CRP	sCD14-ST (Presepsin)	sCD14-ST PCT CRP	0.954 0.847 0.859
Aksaray et al. [13]	n = 90 (52 sepsis; 38 SIRS)	Differentiation Sepsis–SIRS	PCT APACHE II	sTREM-1	sTREM-1 PCT APACHE II	0.78 0.65 0.71
Brenner et al. [14]	n = 90 (60 septic shock; 30 healthy controls)	Septic shock	PCT IL-6 CRP	sTREM-1	sTREM-1 IL-6 PCT CRP	0.955 0.898 0.844 0.791
Khater et al. [15]	n = 80 (40 sepsis; 40 healthy controls)	Sepsis	Lactate	suPAR	suPAR Lactate	0.99 0.84
Yin et al. [16]	n = 171 (151 sepsis; 20 healthy controls)	Sepsis	PCT CRP SOFA	CD64	CD64 PCT SOFA CRP	0.879 0.868 0.701 0.609
Larsson et al. [17]	n = 271 (77 sepsis; 194 non-sepsis)	Sepsis	PCT	Calprotectin	CaPT PCT	0.67 0.55
Spoto et al. [18]	n = 159 (109 sepsis; 50 healthy controls)	Sepsis	PCT SOFA	MR-proADM	MR-proADM PCT SOFA	0.817 0.884 0.774
Hamed et al. [19]	n = 290 (213 sepsis; 77 healthy controls)	Sepsis	PCT IL-6 CRP	PTX-3	PTX-3 PCT IL-6 CRP	0.92 0.92 0.91 0.82
Casagranda et al. [20]	n = 130 (Sepsis)	28-day mortality	Lactate	suPAR	suPAR Lactate	0.77 0.70
Chen et al. [21]	n = 66 (25 septic shock; 11 sepsis; 30 healthy controls)	28-day mortality	APACHE II	HMGB-1	HMGB-1 IL-10 APACHE II	0.946 0.877 0.846
Andaluz-Ojeda et al. [22]	n = 326 (Sepsis)	28-day mortality	PCT CRP Lactate SOFA	MR-proADM	MR-proADM SOFA Lactate PCT CRP	0.79 0.75 0.71 0.61 0.54
Kim H et al. [23]	n = 215 (Sepsis)	30-day mortality	SOFA	Bio-ADM	Bio-ADM SOFA	0.827 0.830
Seol et al. [24]	n = 145 (Sepsis)	28-day mortality	SOFA	Angiopoietin	SOFA Angpt2/1 ratio	0.745 0.736
Fang et al. [25]	n = 388 (333 sepsis; 55 healthy controls)	28-day mortality	PCT	Angiopoietin	Angpt2/1 ratio PCT	0.845 0.732

## Data Availability

Not applicable.

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
