# Peer review of "Cytokine Storm—Definition, Causes, and Implications"

_ijms, 2022, doi:10.3390/ijms231911740_

Round 1
Reviewer 1 Report
In this manuscript, Jarczak and Nierhaus reviewed the very important clinical phenomenon termed cytokine storm (CS) in multiple diseases, with a focus on its pathophysiology, clinical aspects, and potential therapeutic approaches. This review also discussed the immune dysregulation after CS, which is of clinical importance.
Overall, this review is well-written and provides important insights for future studies that focus on the pathogenesis and clinical aspects of the cytokine storm. To strengthen the manuscript, the authors should make efforts to address the following concerns.
Comments:
1. Table 2 presents the diagnostic and prognostic biomarkers for sepsis, which is supposed to support the idea that “the quantitative values of CS biomarkers could not indicate clinical features and prognosis”. Therefore, the authors should highlight the common CS biomarkers (e.g., IL6, CRP, PCT, etc.) in the table and compare their capacities in indicating disease features and prognosis with the clinically used ones (e.g., sCD14-ST, sTREM-1, etc.).
2. The authors used both “hypercytokinemia” and “CS” in the subtitles. Since this review is discussing CS and “hypercytokinemia” is a synonym for CS, it might be better to keep the term consistent throughout the manuscript.
3. In line 164, authors should have citations to support the claim that the short half-lives of cytokines can prevent remote effects, as many cytokine effects are systematic.
4. Several pages are mislabeled starting from line 133.
5. “500.000” in line 433 should be “500,000”
Author Response
REVIEWER 1
Suggestions for Authors
In this manuscript, Jarczak and Nierhaus reviewed the very important clinical phenomenon termed cytokine storm (CS) in multiple diseases, with a focus on its pathophysiology, clinical aspects, and potential therapeutic approaches. This review also discussed the immune dysregulation after CS, which is of clinical importance.
Overall, this review is well-written and provides important insights for future studies that focus on the pathogenesis and clinical aspects of the cytokine storm. To strengthen the manuscript, the authors should make efforts to address the following concerns.
Comments:
- Table 2 presents the diagnostic and prognostic biomarkers for sepsis, which is supposed to support the idea that “the quantitative values of CS biomarkers could not indicate clinical features and prognosis”. Therefore, the authors should highlight the common CS biomarkers (e.g., IL6, CRP, PCT, etc.) in the table and compare their capacities in indicating disease features and prognosis with the clinically used ones (e.g., sCD14-ST, sTREM-1, etc.).
We thank the reviewer for this suggestion. We have modified the table accordingly for a comparison of common markers with less common mediators and parameters (Table 2.)
- The authors used both “hypercytokinemia” and “CS” in the subtitles. Since this review is discussing CS and “hypercytokinemia” is a synonym for CS, it might be better to keep the term consistent throughout the manuscript.
We completely agree. We have changed “hypercytokinemia” to CS throughout the manuscript.
- In line 164, authors should have citations to support the claim that the short half-lives of cytokines can prevent remote effects, as many cytokine effects are systematic.
Thank you. This claim is now supported by its correct reference.
- Several pages are mislabeled starting from line 133.
We apologize for this confusing issue. We were informed that this is the result of the preliminary layout provided by the publisher. The problem will be resolved once typesetting is completed.
- “500.000” in line 433 should be “500,000”
Thank you. We have corrected to 500,000.
Reviewer 2 Report
The review on cytokine storm (CS) by Jarczak and Nierhaus provides a decent summary about the phenomenon. Their efforts in understanding CS in the context of various pathophysiological conditions is appreciable. The review reads well and is definitely informative. However, the review does not appear to provide new ideas or directions (at least these aspects are not easily identifiable) as I would expect from a review article.
I have a few not-so-serious comments outlined below:
1. The title, apparently, is fine, but to me, appears to send a message that the scope of the review would be restricted to only defining CS and nothing beyond. Consequently, I suggest that the authors consider another title, maybe something like: Cytokine storm- causes and implications.
2. Since CS is a very complex phenomenon, and since the goal of the authors and others in this field is to understand the phenomenon in biological and practical terms, I feel that the authors could consider doing the following: Side-by-side compare clinical, epidemiological, biological and clinical aspects (therapeutic approaches and clinical success/failure) of CS between young and old patients, and between viral and non-viral CS.
3. I am personally certain that genetics (GAWS/germline markers) plays a role in developing (and other aspects) CS. I feel that a separate section should be considered discussing potential underlying genetic of CS.
4. A review, in my view, is a treatise that not only summarizes relevant information about a topic but also synthesizes a common reasonable theme/hypothesis and dares to propose what could be/needs to be done in the future. I suggest the reviewers think along this line and make the review a ‘place’ where readers will not only get information but will also get ideas as to what needs to be/could be potentially done in the future (consider adding a section on ‘Future Directions’).
5. In the summary, in the end, use bullet points to highlight important points about CS that will immediately provide relevant information about CS and help the review ‘stand out in the middle of the crowd’.
Author Response
REVIEWER 2
Overall the review is well written and the authors included many interesting topics. Images and tables are well crafted, easy to read, and full of useful information.
However, on the whole, the manuscript is quite long and sometimes difficult to read and to understand.
Criticisms:
- Introduction. I recommend to summarize some parts in order to make it easier to read and to understand the whole topic.
We thank the reviewer for this important suggestion. We have substantially shortened the text and rephrased unclear statements.
- In paragraph n.3 the parts explaining well-known and vast topics are too detailed (e.g. on the complement system, TLRs or PAMPs, the coagulation cascade) and should be eliminated (a reference to previous reviews on those topics can be useful instead).
We completely agree. The third paragraph has been shortened as requested
3 At the end of each paragraph there must be short a sentence explaining the main idea of the whole paragraph and connecting each paragraph with the next one. Currently the length of the individual paragraphs makes the topics sometimes rather confusing and difficult to follow.
Thank you for this suggestion. Below each paragraph, we have added a short summary.
- Paragraphs from 4 to 10 should be written as subsections of a single general paragraph in which the authors should discuss about the conditions that cause a cytokine cascade. Each of these subparagraphs should be reviewed in the light of all the information that was already mentioned throughout the text (and therefore should be shortened) in order to make the reading more fluid.
We completely agree. To accommodate this request, we have changed paragraph 4 (now: “unleashing the cytokine cascade”) and created new subparagraphs accordingly.
Reviewer 3 Report
Overall the review is well written and the authors included many interesting topics. Images and tables are well crafted, easy to read, and full of useful information.
However, on the whole, the manuscript is quite long and sometimes difficult to read and to understand.
Criticisms:
1. Introduction. I recommend to summarize some parts in order to make it easier to read and to understand the whole topic.
2. In paragraph n.3 the parts explaining well-known and vast topics are too detailed (e.g. on the complement system, TLRs or PAMPs, the coagulation cascade) and should be eliminated (a reference to previous reviews on those topics can be useful instead).
3 At the end of each paragraph there must be short a sentence explaining the main idea of the whole paragraph and connecting each paragraph with the next one. Currently the length of the individual paragraphs makes the topics sometimes rather confusing and difficult to follow.
4. Paragraphs from 4 to 10 should be written as subsections of a single general paragraph in which the authors should discuss about the conditions that cause a cytokine cascade. Each of these subparagraphs should be reviewed in the light of all the information that was already mentioned throughout the text (and therefore should be shortened) in order to make the reading more fluid.
Author Response
REVIEWER 3
The review on cytokine storm (CS) by Jarczak and Nierhaus provides a decent summary about the phenomenon. Their efforts in understanding CS in the context of various pathophysiological conditions is appreciable. The review reads well and is definitely informative. However, the review does not appear to provide new ideas or directions (at least these aspects are not easily identifiable) as I would expect from a review article.
I have a few not-so-serious comments outlined below:
- The title, apparently, is fine, but to me, appears to send a message that the scope of the review would be restricted to only defining CS and nothing beyond. Consequently, I suggest that the authors consider another title, maybe something like: Cytokine storm- causes and implications.
Thank you for this important comment. We have changed the title as suggested.
- Since CS is a very complex phenomenon, and since the goal of the authors and others in this field is to understand the phenomenon in biological and practical terms, I feel that the authors could consider doing the following: Side-by-side compare clinical, epidemiological, biological and clinical aspects (therapeutic approaches and clinical success/failure) of CS between young and old patients, and between viral and non-viral CS.
We totally agree. As suggested, we now inserted paragraph 6 “viral vs. non-viral CS” and paragraph 7 (age-related changes of the immune system”
- I am personally certain that genetics (GAWS/germline markers) plays a role in developing (and other aspects) CS. I feel that a separate section should be considered discussing potential underlying genetic of CS.
Thank you. We have added paragraph 8 “Endotypes, GWAS, and transcriptome analysis” to accommodate this request.
- A review, in my view, is a treatise that not only summarizes relevant information about a topic but also synthesizes a common reasonable theme/hypothesis and dares to propose what could be/needs to be done in the future. I suggest the reviewers think along this line and make the review a ‘place’ where readers will not only get information but will also get ideas as to what needs to be/could be potentially done in the future (consider adding a section on ‘Future Directions’).
We completely agree and have added paragraph 9 “Future perspectives”.
- In the summary, in the end, use bullet points to highlight important points about CS that will immediately provide relevant information about CS and help the review ‘stand out in the middle of the crowd’.
Thank you for this important suggestion. We have added “take home messages” using bullet points.